## RESEARCH ARTICLE

# Giving their all for their offspring: physiological trade-offs in an Andean-Patagonian viviparous lizard in response to global warming

Jimena B. Fernández*, Erika L. Kubisch, Fernando Duran and Jorgelina M. Boretto

## ABSTRACT

Global warming threatens biodiversity, particularly affecting ectothermic animals, which must seek refuge to avoid overheating when ambient temperatures exceed their critical thresholds. Extended shelter use limits the time for essential activities such as foraging, social interactions, and reproduction, potentially reducing survival and increasing local extinction risk. Viviparous Liolaemids inhabiting cold-temperate Andean regions are considered vulnerable to rising temperatures and are predicted to experience local extinctions this century. We evaluated the effects of thermal restriction on pregnancy outcomes and offspring in the viviparous lizard *Liolaemus pictus* under two conditions. One group of pregnant females experienced simulated future thermal restrictions (restriction group, RG; $n$=12), while another group experienced identical laboratory conditions without thermal restrictions (no-restriction group, NRG; $n$=14). In RG females, 41.7% were removed due to feeding cessation or spontaneous abortions (versus 12.3% in NRG), reflecting the consequences of thermal restriction. The remaining RG females selected lower preferred body temperatures after 2 weeks of experimentation and maintained a stable body condition throughout pregnancy. However, both groups of offspring showed similar body condition and locomotor performance, suggesting physiological compensation by RG females. This physiological plasticity of *L. pictus* may help buffer the adverse effects of global warming on reproductive success.

KEY WORDS: Climate change, Hours of restriction, Physiological plasticity, Pregnant females, *Liolaemus pictus*

## INTRODUCTION

Global warming driven by anthropogenic activities is altering climate conditions worldwide (IPCC AR6 WGI, 2021; IPCC AR6, 2023). In Argentina, this has become evident over the past few decades, with record-high temperatures and linear trends toward drier conditions in hydrological variables in the Andes and Patagonia, such as precipitation decline, decrease in rivers'

streamflow, and less snow cover; leading to glaciers receding and a rise in wildfire events (Villalba et al., 2005; Veblen et al., 2011; Pessacg et al., 2020; Kitzberger et al., 2022; Hurtado et al., 2023). Projections for northwestern Patagonia suggest an increase in mean near-surface air temperature between 3 and 3.5°C and a decrease in precipitation between 0.75 and 1 mm per day (under an extreme emissions scenario, RCP "Representative Concentration Pathway", scenario 8.5) for the next 50 years compared to current conditions (IPCC AR5, 2014; IPCC AR6, 2023). This warming is expected to be more intense during summer months (Barros et al., 2015; IPCC AR6, 2023). Projected temperature increases over the coming decades in Argentina are greater than the warming observed over the past 60 years, but this will occur in half the time, representing a regional warming acceleration that is five times greater than that experienced during the last century (Meehl et al., 2007; Barros et al., 2015).

One of the greatest challenges facing biology today is to infer the impact of climate change on organisms and ecosystems to develop actions that allow for the mitigation and conservation of species (Allen et al., 2010; Sinclair et al., 2016; Kacoliris et al., 2020; Miles, 2020; Habibullah et al., 2022). Ectotherms are particularly valuable models for predicting responses to abiotic changes, as their physiological processes are tightly regulated by ambient temperature (Deutsch et al., 2008; Angilletta, 2009; Winter et al., 2016; Wild et al., 2025). In particular, the increase in ambient temperature due to climate change is affecting various aspects of the life history of reptiles, such as embryonic development, growth rates, locomotion, dispersal, reproduction, and even individual behavior (Buckey and Huey, 2016; Sinclair et al., 2016; Ibargüengoytía et al., 2020; Zhang et al., 2022; Li et al., 2024).

Organisms respond to new environmental conditions through two main pathways: via phenotypic plasticity, by modifying their behavior and physiology, or, over the longer term, by evolving new genetic adaptations. However, the current pace of climate change is too fast for the latter possibility to be a viable option for many species (Deutsch et al., 2008; Roulin, 2014; Bezeng et al., 2018; Wild and Gienger, 2018). A recent study suggests that species will have limited capacity to respond to the unprecedented rapid global warming, as heat tolerance evolves more slowly than cold tolerance in both endotherms and ectotherms (Bennett et al., 2021). In particular, reptiles can compensate, to some extent, for variations in ambient temperature and maintain a relatively constant and appropriate body temperature through physiological and behavioral thermoregulation (Angilletta, 2009). However, this efficiency will depend on the availability of sufficient thermal heterogeneity in the environment and the individual's phenotype (Huey et al., 2003; Clusella-Trullas et al., 2011; Rutschmann et al., 2020; reviewed for Liolaemidae in Cruz et al., 2021). Therefore, when ambient temperatures rise beyond the optimal levels for metabolic processes, exceeding the 'physiological

Laboratorio de Ecofisiología e Historia de vida de Reptiles, Instituto de Investigaciones en Biodiversidad y Medio Ambiente (INIBIOMA), Consejo Nacional de Investigaciones Científicas y Técnicas (CONICET) - Centro Regional Universitario Bariloche, Universidad Nacional del Comahue, 8400 San Carlos de Bariloche, Río Negro, Argentina.

*Author for correspondence ( jimenafernandez@comahue-conicet.gob.ar)

(ID) J.B.F., 0000-0002-2688-2748; E.L.K., 0000-0002-1344-4052; F.D., 0000-0002-7256-1478; J.M.B., 0000-0002-3531-0442

Biology Open

limits' of heat tolerance (Bennett et al., 2021), lizards seek refuge to avoid overheating, which leads to a restriction in activity time. These hours of restriction ($h_r$), in which lizards are forced into shelters due to temperatures exceeding their thermal preferences (*sensu* Sinervo et al., 2010), may lead to reduced foraging, social interactions, and reproductive activities, thereby increasing the risk of local extinctions (Sinervo et al., 2010; Kubisch et al., 2016a; Vicenzi et al., 2017; see review in Ibargüengoytía et al., 2020).

In viviparous lizard species in particular, temperature restrictions during activity time in the reproductive season can compromise reproductive performance (mating and embryonic development) and lead to local extinctions (Sinervo et al., 2010; Jara et al., 2019). This likely relates to the varying thermal requirements observed among females based on their reproductive status. While non-pregnant females typically select a broad range of body temperatures (Fernández et al., 2017), pregnant females tend to prefer more stable temperatures to ensure optimal conditions for embryonic development (Ji et al., 2007; Li et al., 2009; Cruz et al., 2014; Fernández et al., 2017; López-Alcaide et al., 2017). Additionally, developing embryos (especially oviparous ones) and juveniles are particularly susceptible to high temperatures, as they have narrower thermal tolerance ranges than adults, and lower thermal inertia (Levy et al., 2015; Buckley and Huey, 2016). Therefore, sustained increases in temperature during the early stages of development can affect the locomotor performance and behavioral patterns of newborns, thereby influencing both short- and long-term biological fitness of the species (Aidam et al., 2013; Angilletta et al., 2013; Buckley and Huey, 2016; Fernández et al., 2017).

According to previous studies, viviparous *Liolaemus* species inhabiting cold-temperate regions of the Andes are at high risk of experiencing population declines, extirpations, or even extinctions within the next half-century (Pincheira-Donoso et al., 2013; Jara et al., 2019). This is because viviparity in this genus appears to be restricted to cold environments due to its evolution as an adaptation to such conditions (Pincheira-Donoso et al., 2013). However, Ibargüengoytía et al. (2021) found that viviparous *Liolaemus* species maintain lower body temperatures in the field compared to oviparous species, despite both reproductive modes exhibiting similarly high preferred temperatures. This key difference may provide a thermal buffer for viviparous species, helping them cope with global warming. The same has been reported in North American *Sceloporus* lizards, which serve as a model system for predicting the impact of climate warming on oviparous and viviparous species (Ma et al., 2022). Nevertheless, empirical studies investigating the potential effects on offspring phenotype of mothers' exposure to $h_r$ due to extremely high ambient temperatures during pregnancy are lacking. Pregnant females exposed to high ambient temperatures may be unable to thermoregulate effectively, potentially failing to maintain optimal body temperatures for embryonic development.

In this study, we conducted laboratory experiments to investigate the potential effects of increased ambient temperature and $h_r$ resulting from projections of climate warming on the reproductive success and offspring phenotype of the viviparous lizard *Liolaemus pictus* Duméril and Bibron (1837). *L. pictus* (Fig. 1) is a characteristic species of the cold-temperate climate found in the northern Andean Patagonian forests of Argentina and Chile. While it is categorized as 'Least Concern' (IUCN; Abdala et al., 2016 in Argentina and Ministry of Environment, MMA, 2014 in Chile) due to its richness in many parts of its range (Ávila et al., 2006), its population abundance in Chile has declined due to habitat destruction and invasive tree species over the past 150 years

(Vera-Escalona et al., 2010). According to the IUCN, the current population trend of *L. pictus* is 'Decreasing' (Abdala et al., 2016), and the extent to which global warming contributes to this decline remains unknown.

We hypothesize that, under a climate change scenario, rising ambient temperatures will extend periods of thermal restriction in lizards, forcing pregnant females to remain in shelters at body temperatures below their thermal preference. This, in turn, will negatively affect pregnancy success by decreasing the number of viable offspring. Furthermore, we predict detrimental effects on embryonic development and offspring fitness, manifested as a decrease in body condition and diminished locomotor performance. This approach will enable us to evaluate how projected global warming may affect not only this species but also the ability of viviparous Liolaemids to adapt and persist in a changing environment.

## RESULTS

### Field data

Regarding field data, the mean body temperature ($T_b$) of pregnant females ($n=26$) of *L. pictus* at capture was $31.34\pm0.40°C$. The mean microenvironmental temperatures of females' capture site were $T_s=26.56\pm0.56°C$ and $T_a=22.93\pm0.50°C$. Pregnant females randomly assigned to the NRG and RG treatments (Fig. 2) did not show differences in the mean snout-vent length (SVL) (NRG=$61.21\pm0.96$ mm, $n=14$; RG=$61.84\pm0.87$ mm, $n=12$; $t_{24}=0.480$, $P=0.636$) nor in the mean body condition index (BCI$_1$; NRG=$0.490\pm0.04$, $n=14$; RG=$0.485\pm0.03$, $n=12$; $t_{24}=-0.319$; $0.753$).

### Ultrasound diagnosis

Ultrasound diagnosis confirmed that all females were in the initial to medium stages of embryonic development (Fig. 3A,B). Twenty days after the start of experiments females had advanced stage of pregnancy (Fig. 3C). The litter size estimated by ultrasound diagnosis for each female before the start of the experiment was higher than the actual number of offspring they finally delivered (mean litter size by ultrasound diagnosis: $4.61\pm0.28$; mean litter size delivered: $3.54\pm0.26$, min=1, max=6; $t_{25}=3.74$; $P<0.001$).

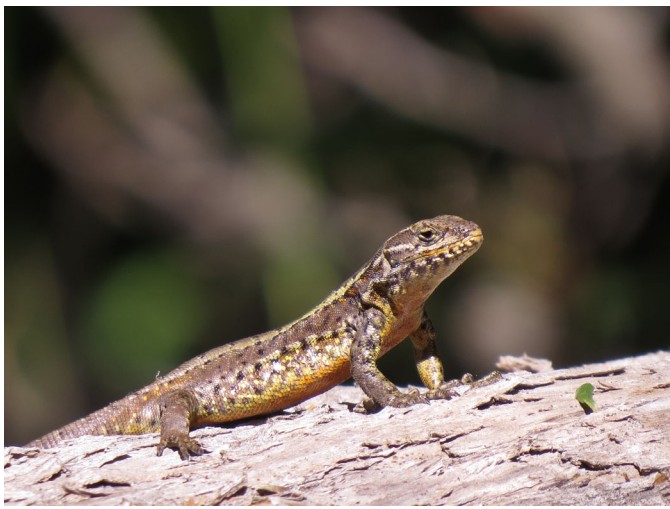

**Fig. 1. Adult female of the viviparous lizard *L. pictus* during pregnancy.** The species was photographed in San Carlos de Bariloche, Argentina, where individuals were captured for this study. Photo credit: E. Kubisch.

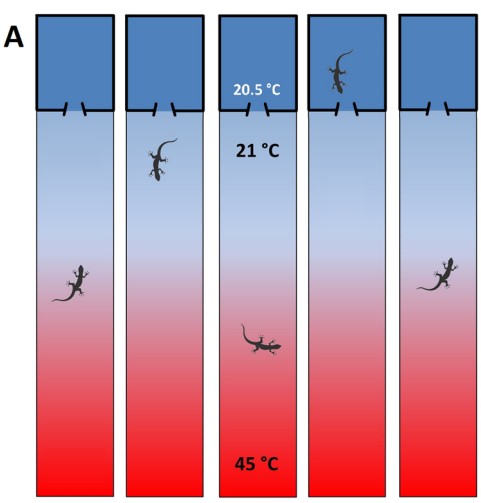

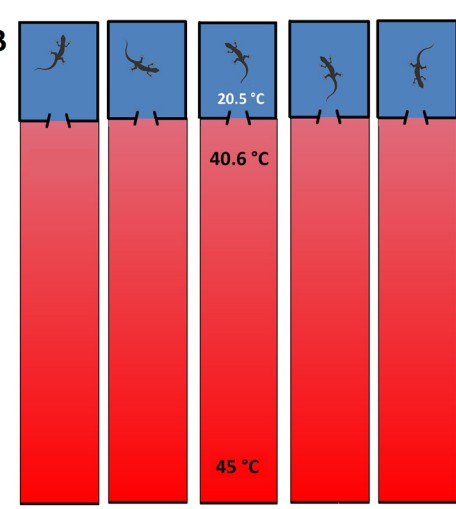

**Fig. 2. Experimental design showing the temperature treatments applied to pregnant *L. pictus* females and their offspring.** (A) NRG. Individuals had free access to a thermoregulatory area with a thermal gradient from 21-45°C (available during activity hours from 9:00 AM to 6:00 PM) and to a constant-temperature shelter at 20.5°C (*n*=14). (B) RG. Same setup as NRG, but during hours of restriction (4.5 h, between 12:00 PM and 4:30 PM), individuals could only choose between exposure to 40.6-45°C or refuge at 20.5°C (*n*=12).

## Pregnant females experiment

The comparative analyses on BCI of pregnant females showed no interaction between group (RG versus NRG) and time (BCI$_1$: at the beginning of temperature treatments versus BCI$_2$: after 15 days under experiment) on BCI (two-way repeated-measures ANOVA: $F_{1,18}$=1.327, $P$=0.264; Fig. 4). Likewise, no main effects were detected for time ($F_{1,18}$=0.145, $P$=0.708) or group ($F_{1,18}$=1.733, $P$=0.205) on BCI. Nevertheless, NRG females showed a tendency to increase their BCI 15 days after the start of the experiment (NRG BCI$_1$=0.490±0.04, BCI$_2$=0.504±0.04, $t_8$=−2.79; $P$=0.023; Fig. 4), whereas in RG females, BCI remained unchanged (RG BCI$_1$=0.485 ±0.03, BCI$_2$=0.486±0.07, $t_{10}$=−0.10; $P$=0.923; Fig. 4).

Four pregnant females from the RG had to be removed from the experiment (between 15 and 22 days under temperature restriction) because they showed weight loss and difficulty in being fed. In addition, one female from the RG and two females from the NRG had spontaneous abortions after 38-39 days under experiment. Thus, more females gave birth and completed the experiment in the NRG treatment (NRG: 12 females out of the initial 14=87.71%; RG: seven females out of the initial 12=58.33%). Binomial test evidence that in the RG the proportion of pregnant females that remained in the experiment until the end was similar to the proportion that had to be removed ($P$=0.581), while in the NRG the proportion of pregnant females that completed the experiment was higher than the proportion that had to be removed ($P$=0.002).

There were no differences in the mean number of days from the start of the experiment to the day of parturition between females from NRG and RG that completed the experiment (NRG=21 days, *n*=12; RG=25 days, *n*=7; $U$=27,000, $P$=0.204). Neither in the average number of viable offspring delivered by females of each treatment group (NRG=3.42, *n*=12; RG=3.71, *n*=7; $t_{17}$=0.397, $P$=0.697). In addition, for females that completed the experiment, no interaction between group (RG versus NRG) and time (BCI$_3$: immediately after parturition versus BCI$_4$: 10 days postpartum) was found on BCI, considering that both NRG and RG were maintained in similar thermal conditions after parturition (two-way repeated-measures ANOVA: $F_{1,16}$=0.224, $P$=0.642). The BCI did not differ between groups ($F_{1,16}$=1.807, $P$=0.198), but increased after 10 days postpartum in both NRG and RG females ($F_{1,16}$=14.645, $P$=0.001).

### $T_{pref}$ of pregnant females

The mean preferred temperature ($T_{pref}$) of pregnant females when they arrived at the laboratory ($T_{pref}$1) was 34.83±0.22°C, and the minimum and maximum temperature set-points ($T_{set}$; as the 25 and 75% interquartile) of the $T_{pref}$ were 33.39±0.46°C and 35.70±0.14°C, respectively. An interaction between group (RG versus NRG) and time ($T_{pref}$1 versus $T_{pref}$ after 15 days under experiment, $T_{pref}$2) was observed on $T_{pref}$ (two-way repeated-measures ANOVA: $F_{1,16}$=5.180, $P$=0.037; Fig. 5). This indicates that the pattern of change in $T_{pref}$ over time differs between the groups. The RG group showed a decrease in $T_{pref}$ from $T_{pref}$1 to $T_{pref}$2, whereas no significant change was found within the NRG group (Table 1, Fig. 5). In addition, there were no differences in $T_{pref}$1 between pregnant females of the NRG and RG treatments, but after 15 days under experiment, the females of the NRG group had a higher $T_{pref}$ than those of the RG group ($T_{pref}$2; Table 1, Fig. 5).

One pregnant female from the NRG and one from the RG gave birth during the $T_{pref}$2 recording. The mean $T_{pref}$ of the RG parturient female was 35.65±0.030°C (*n*=901) and the mean $T_{pref}$ of the NRG parturient female was 34.65±0.032°C (*n*=2522).

### Reproductive effort, body condition of newborns, and morphometric measures

There was no difference between the mean reproductive effort of females of both treatments (RG=0.079, *n*=7; NRG=0.083, *n*=11, $t_{16}$=−0.543, $P$=0.595). In addition, an interaction between group (RG versus NRG) and time (BCI$_{birth}$: at birth versus BCI$_{10-d}$: after 10 days from birth, being under the same temperature treatment as their mothers before parturition) was observed on the offspring's BCI (two-way repeated-measures ANOVA: $F_{1,63}$=4.089, $P$=0.047). This indicates that the pattern of change in BCI over time differs between the groups. The newborns' BCI$_{10-d}$ of NRG decreased while the BCI$_{10-d}$ of RG's newborns remained similar to that at birth (BCI$_{birth}$, Table 2). Furthermore, the BCI$_{birth}$ and BCI$_{10-d}$ did not differ between NRG and RG (Table 2). Also, no differences were found in any of the morphometric variables measured in newborns between the NRG and RG treatments (Table 3).

### Newborns' locomotor performance

There were no differences between the $V_{max}$ of NRG and RG newborns in any run (sprint run, SR; or long run, LR), or at any run temperature (21°C or 34°C), either at birth or after 10 days (Table S1; Fig. 6). Newborns of NRG and RG run faster at 34°C than at 21°C, both in SR and LR, at birth and also 10 days after (Table S2; Fig. 6). Newborns of NRG at birth run faster in SR than LR at 21°C; while newborns of RG 10 days after birth run faster in

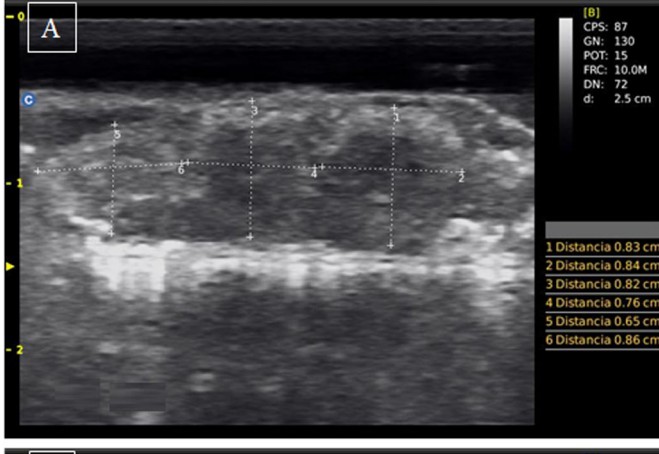

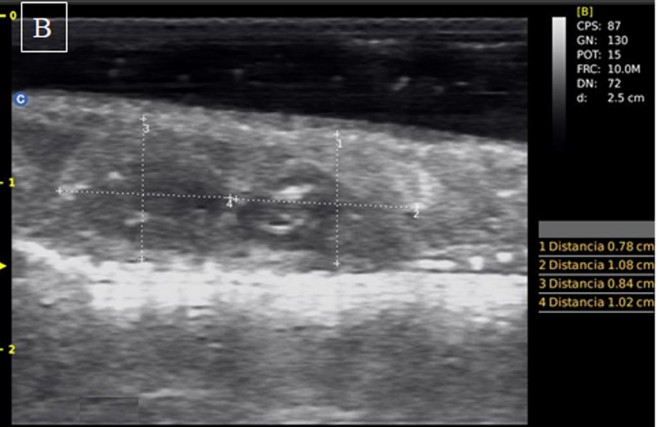

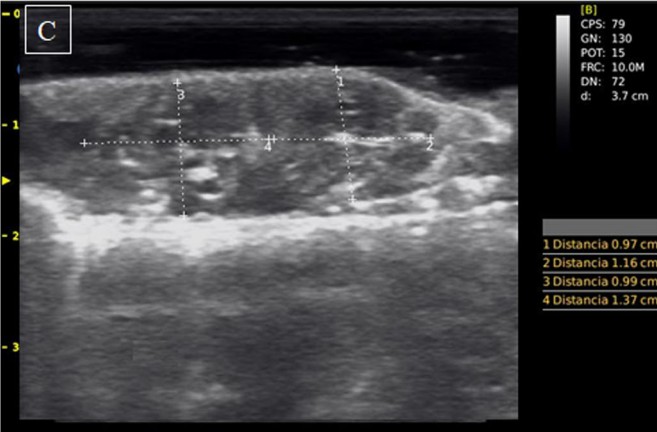

**Fig. 3. Ultrasound images showing different stages of pregnancy in _L. pictus_ females.** Images correspond to the left uterus at successive stages of pregnancy (classified according to Leyton et al., 1980), from the beginning of the experiments to 20 days later: (A) initial, (B) medium, and (C) advanced pregnancy. Embryos are indicated with measured lines.

LR than SR at 21°C; and, in addition, they run faster at birth than 10 days after in SR at 21°C (Table S2; Fig. 6).

## DISCUSSION

Globally, reptiles are a key animal group for inferring the potential effects of projected increases in ambient temperatures in the coming years (Ibargüengoytía et al., 2020; Li et al., 2024; Wild et al., 2025). In this study, a widely distributed viviparous lizard from northwestern Patagonia, where the climate is cold-temperate, was used as a model to evaluate the potential effects of an extreme global

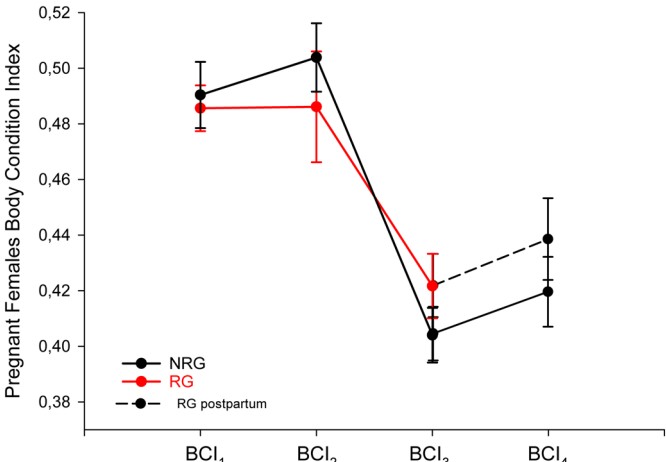

**Fig. 4. Body condition index of pregnant _L. pictus_ females across experimental stages.** Means±s.e.m. values are shown for the NRG (black line; _n_=14) and the RG (red line; _n_=12). Measurements were taken at four time points: at the start of the experiment (BCI$_1$), after 15 days under the temperature treatments (BCI$_2$), immediately after parturition (BCI$_3$), and 10 days postpartum (BCI$_4$; RG postpartum, dashed line).

warming scenario on pregnant females and their offspring. Contrary to our predictions, the $h_r$ due to extremely high ambient temperatures in pregnant females of _L. pictus_ did not show short-term negative effects on pregnancy success, compared to females able to reach their preferred temperatures during all active hours. Furthermore, no differences were observed between treatment groups (NRG and RG) in the number of viable offspring, body condition of newborns, and newborns' locomotor performance at birth, as well as after 10 days under the same temperature restriction treatments as their mothers. However, 41.7% of RG females had to be removed from the experiment due to cessation of feeding or spontaneous abortions, in contrast to the 12.3% of NRG females, reflecting not only the impact of thermal restriction on pregnancy viability in the first group, but also the effects of captivity-induced stress in both groups.

In addition, the results showed that females subjected to thermal restriction during pregnancy did not increase their body condition,

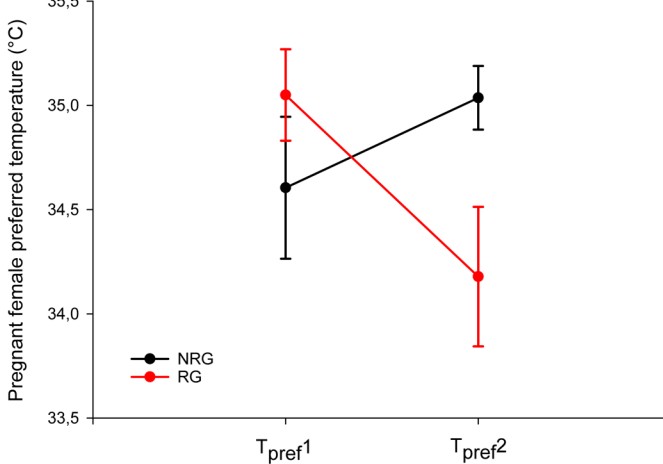

**Fig. 5. Preferred body temperature ($T_{pref}$) of pregnant _L. pictus_ females under experimental conditions.** Means±s.e.m. values are shown for the NRG (black line; _n_=14) and the RG (red line; _n_=12). Measurements were taken at two time points: at the start of the experiment ($T_{pref}$1) and after 15 days under the temperature treatments ($T_{pref}$2).

**Table 1. Pregnant females' preferred body temperature after the beginning of experiments ($T_{pref}1$) and after 15 days under temperature treatments ($T_{pref}2$)**

| Temperature treatments and statistical tests | $T_{pref}1$ (°C) | $T_{pref}2$ (°C) | Paired $t$-test; $P$ |
|---|---|---|---|
| NRG | 34.60±0.76 | 35.04±0.48 | $t_9$=−0.38, 0.715 |
| RG | 35.12±0.34 | 34.18±0.95 | $t_7$=2.85, 0.025* |
| $t$-test; $P$ | $t_{23}$=1.18; 0.250 | $t_{16}$=−2.50; 0.024* | |

Values are means±s.e.m. resulting from the comparative analysis between and within temperature treatments of NRG and RG. $t$, $t$-statistic with degrees of freedom; $P$, $P$-value ($\alpha$=0.05). Asterisks indicate a significant difference (*$P$<0.05).

unlike NRG pregnant females, which tended to do so. This pattern likely reflects the allocation of most metabolic energy by thermally restricted females toward maximizing offspring fitness at birth. These females appeared to compensate physiologically for the thermal restrictions, exhibiting a reproductive effort similar to that of NRG females and producing offspring with body condition and locomotor performance comparable to those of females that experienced no thermal restriction during pregnancy. These results are in contrast with the pattern reported for the viviparous lizard *Carinascincus metallicus* under restricted basking opportunities (Swain and Jones, 2000). Notably, the body condition of NRG newborns declined within 10 days after birth, whereas in RG newborns it remained similar to that at birth. We hypothesize that pregnant *L. pictus* females exposed to thermal stress may allocate more energy to their offspring, enabling them to withstand postnatal thermal challenges and potentially improving survival in a thermally stressful and, consequently, activity-restricted environment. Previous studies in other Liolaemid species, such as in *Liolaemus elongatus* (Halloy et al., 2007), *Phymaturus punae* (Boretto et al., 2007), *Phymaturus zapalensis* (Boretto and Ibargüengoytía, 2009), *Phymaturus spectabilis* (Boretto et al., 2014), *Phymaturus aguanegra* (Cartes et al., 2010), and *Phymaturus antofagastensis* (Boretto et al., 2018), have documented high maternal energetic allocation to newborns, which are born with abundant fat reserves and large amounts of intra-abdominal yolk. Such reserves could enhance offspring survival even when parental investment negatively affects future maternal reproduction (Boretto et al., 2007, 2014; Boretto and Ibargüengoytía, 2009; Cartes et al., 2010). Future studies could determine whether *L. pictus* females also allocate energy to their litter in the form of intra-abdominal yolk and whether the magnitude of this reserve differs between offspring from mothers exposed or not to thermal restriction during pregnancy.

Regarding thermal selection, despite the consideration of $T_{pref}$ as a conservative and low-variation parameter for genus *Liolaemus*

**Table 2. Newborns' body condition index at birth ($BCI_{birth}$) and after 10 days being under the same temperature treatment as their mothers ($BCI_{10-d}$)**

| Temperature treatments and statistical tests | $BCI_{birth}$ ($n$) | $BCI_{10-d}$ ($n$) | Paired $t$-test; $P$ |
|---|---|---|---|
| NRG | −0.263±0.061 (42) | −0.286±0.095 (39) | $t_{38}$=3.193; 0.003* |
| RG | −0.261±0.067 (28) | −0.257±0.114 (26) | $t_{25}$=−0.324; 0.749 |
| $t$-test; $P$ | $t_{68}$=0.114; 0.909 | $t_{63}$=1.132; 0.262 | |

Values are means±s.e.m. resulting from the comparative analysis between and within temperature treatments of NRG and RG. $t$, $t$-statistic with degrees of freedom; $P$, $P$-value ($\alpha$=0.05). Asterisks indicate a significant difference (*$P$<0.05).

**Table 3. Newborns' morphometric variables between temperature treatments of the NRG and the RG**

| Morphometric variables (mm) | Treatments (mean or median) | | Statistical test (Mann–Whitney or $t$-test; $P$) |
|---|---|---|---|
| | NRG ($n$) | RG ($n$) | |
| SVL | 25.31, 24.51-26.58 (41) | 25.66, 24.99-27.18 (26) | $U$=453.000; 0.309 |
| Head length | 7.90±0.59 (40) | 8.05±0.82 (26) | $t_{64}$=−0.863; 0.391 |
| Head width | 5.22±0.31 (40) | 5.36±0.45 (26) | $t_{64}$=−1.480; 0.144 |
| Inter-limb length | 11.74±1.23 (39) | 11.92±1.61 (26) | $t_{63}$=−0.506; 0.615 |
| Distance between knees | 10.59±0.77 (40) | 10.85±1.20 (26) | $t_{64}$=−1.704; 0.287 |
| Tail length | 38.77, 33.01-41.75 (40) | 37.62, 33.33-41.25 (26) | $U$=484.000; 0.641 |

Values are means±s.e.m. or median (max-min) resulting from the comparative analysis between morphometric variables of NRG and RG. SVL, snout-vent length; $U$, $U$-statistic; $t$, $t$-statistic with degrees of freedom; $P$, $P$-value ($\alpha$=0.05).

(Medina et al., 2009; see review of Cruz et al., 2021), pregnant *L. pictus* exhibited a lower mean $T_{pref}$ before the start of the experiments (34.8°C) compared to those previously reported for females and males (36.2°C; Gutiérrez et al., 2010) of the same species. This could be related to the physiological need to maintain low and/or stable temperatures during pregnancy, as has also been observed in its congener *Liolaemus sarmientoi* (Fernández et al., 2015), and other lizard species (e.g. *Uta stansburiana*, Paranjpe et al., 2013; *Amalosia lesueurii*, Dayananda et al., 2017; *Sceloporus* spp., López-Alcaide et al., 2017). In addition, following 15 days under experimental conditions, pregnant *L. pictus* females subjected to thermal restriction exhibited physiological compensation that appears to influence their selection of lower $T_{pref}$, likely as a strategy to maintain offspring fitness stability while maintaining stable body condition. In contrast, NRG females sustained unchanged $T_{pref}$ throughout pregnancy. Similarly, pregnant females of the viviparous skink *Saiphos equalis* have been shown to compensate for elevated temperatures during pregnancy by modifying their thermoregulatory and foraging behaviors, which incurs an energetic cost manifested in reduced maternal body condition (Beltrán et al., 2021). However, as in *L. pictus*, no differences were observed in the morphology, thermal preference, or growth rate of *S. equalis* offspring. Nonetheless, mothers produced offspring with reduced foraging and locomotor performance, suggesting that in this species, the protective effect against elevated developmental temperatures may be incomplete (Beltrán et al., 2021). Comparable behavioral shifts in response to high temperatures have also been reported in oviparous lizards, such as changes in nest site selection to avoid locations with high incubation temperatures [e.g. *Acritoscincus duperreyi* (formerly *Bassiana duperreyi*), Shine and Harlow, 1996; *Intellagama lesueurii* (formerly *Physignathus lesueurii)*, Doody et al., 2006; *Anolis sagrei*, Pruett et al., 2020; *Oedura lesueurii*, Dayananda et al., 2016].

Following parturition, RG females maintained under the same conditions as NRG females exhibited significant mass recovery, resulting in improved body condition. It is likely that, if RG females continued to experience hours of activity restriction due to high temperatures after parturition, they would have died or, at the very least, would have failed to recover their body condition sufficiently for safe release into their natural habitat. This possibility could be evaluated in future studies. In this context, the physiological compensation observed in *L. pictus* may represent an important form of plasticity, particularly in years when heat waves force females to sacrifice their body condition to ensure offspring viability. However,

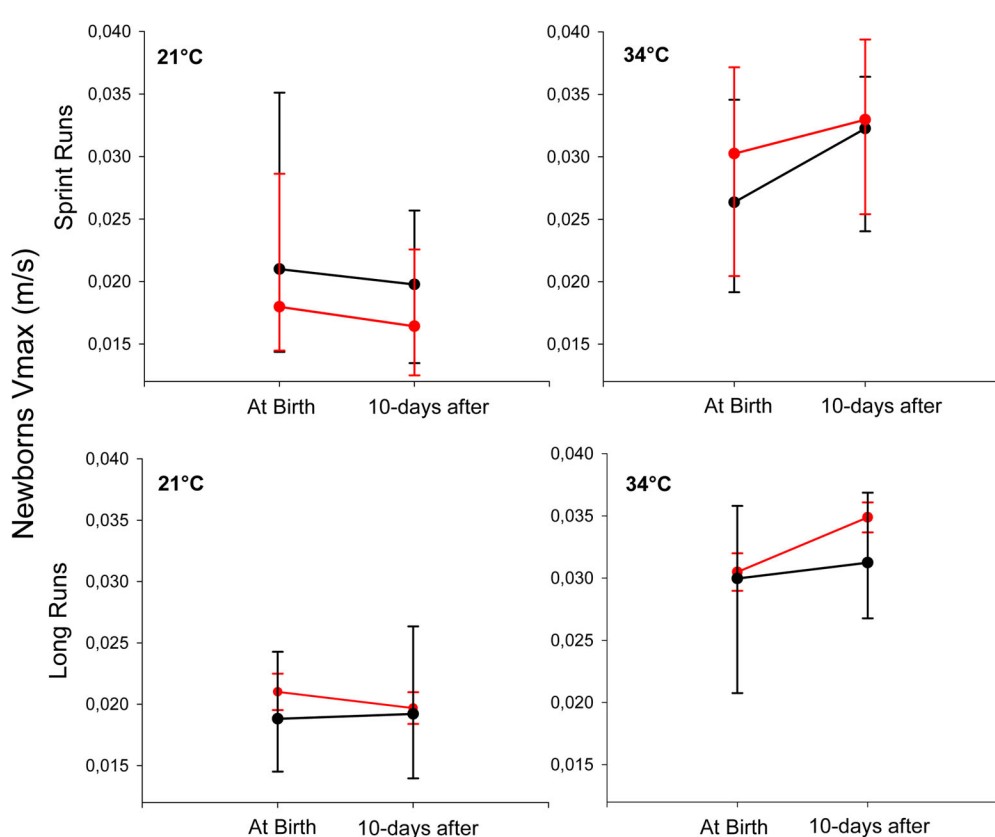

**Fig. 6. Locomotor performance of *L. pictus* newborns from experimental treatments.** Means ±s.e.m. values of maximum running speed ($V_{max}$) are shown for the NRG (black line; $n$=40) and the RG (red line; $n$=26). Performance was evaluated in sprint and long runs at two test temperatures (21°C and 34°C), both at birth and 10 days later.

if such conditions occur annually, females may lack the reserves or the capacity to replenish them sufficiently to survive the winter and initiate a new vitellogenic cycle. Consequently, reproductive females might skip more breeding seasons, potentially becoming triennial breeders. Such a shift could ultimately impact population dynamics and the species as a whole, with cascading ecological consequences under rising ambient temperatures (Deutsch et al., 2008; Allen et al., 2010; Cahill et al., 2013).

In this study, we documented the detrimental consequences that hours of restriction may have during pregnancy under an extreme global warming scenario, since four pregnant females from the RG had to be removed from the experiment because they showed weight loss and difficulty in being fed. In the RG group, 58% of pregnant females completed the experiment, compared to 88% in the NRG group. These results evidence that not all females can successfully deal with this physiological challenge. Our research efforts are future-focused on determining the consequences of chronic warming on the reproductive times of *L. pictus* females in a long-term study. The goal will be to identify the costs of repeated reproductive compensations under chronic warming in terms of modifications of the biannual-triennial cycle that females performed 30 years ago (described by Ibargüengoytía and Cussac in 1996).

Experimental studies such as the one presented here are a valuable proxy for determining the consequences of global warming on animal physiology. In this work, we complemented the use of a non-lethal technique, the ultrasound diagnosis, which is not widely used in wild reptiles in general, and in Argentine lizards in particular. This diagnostic tool enabled us to experimentally evaluate wild females with confirmed pregnancies, and to adjust the start time of the study so that experiments could begin with females already in early to mid-pregnancy. In this study, assessing litter size using ultrasound diagnostics appears to overestimate the data,

indicating the need to continue refining this diagnostic technique, which is complex due to the small size of the species we work with. Nonetheless, this may also reflect abortive or resorptive processes occurring post hoc in some pregnant females, a phenomenon observed in the viviparous lizards *Carinascincus ocellatus* (formerly *Niveoscincus ocellatus*; Jones et al., 1997) and also suggested for the species *Carinascincus metallicus* (Swain and Jones, 2000), both from Tasmania, indicating the need for further investigation into this issue.

Overall, this study highlights the role that the physiological plasticity in pregnant females of a cold-temperate Liolaemid lizard may play in mitigating the effects of global warming. In this context, the impact of high environmental temperatures on the reproductive behavior and physiology of ectotherms could be assessed through studies like this one. However, these effects could be more severe if the restriction occurs naturally and persistently year after year. In the case of *L. pictus*, previous studies have demonstrated their vulnerability to rising temperatures under climate change models (Kubisch et al., 2016a) and their limited capacity to acclimate to new thermal conditions (Kubisch et al., 2016b). Therefore, an integrated study of *L. pictus* focusing on both physiological and behavioral responses to rising ambient temperatures will help elucidate how projected increases in environmental temperature may affect their fitness. Additionally, this approach will enable researchers to determine whether the species possesses behavioral and physiological traits that confer greater resilience to cope with climate change.

## MATERIALS AND METHODS
### Model species
*Liolaemus pictus* (Fig. 1) is a medium-sized lizard species (maximum SVL: 72.3 mm; Abdala et al., 2021) with insectivorous and frugivorous feeding

habits, acting as an important disperser of viable seeds (Wilson et al., 1996; Vidal and Sabat, 2010). Males, unlike females, possess orange pre-cloacal pores, larger head length and width, and a wider tail base width; the latter being the most reliable field trait for sex identification prior to capture (*sensu* Ibargüengoytía and Cussac, 1996). This species is viviparous, reaches sexual maturity at 49 mm of SVL, and females exhibit prolonged reproductive cycles (biennial or triennial), with litter sizes ranging from three to six offspring (Ibargüengoytía and Cussac, 1996). Physiological thermal studies evidence that *L. pictus* achieves optimal locomotor performance within its preferred temperature range, currently available in its environment (Kubisch et al., 2011), and it also displays limited capacity for acclimatization to novel thermal conditions (Kubisch et al., 2016b). Climate projections suggest that, if global temperature continues to rise, *L. pictus* could face up to a 15% risk of local extinction by 2080 (according to IPCC, 2014, climate projection IV scenario A2; Kubisch et al., 2016a), highlighting its vulnerability to rising temperatures (Kubisch et al., 2016a, b). However, these risks have not been assessed at the level of embryonic development and physiological performance of the offspring.

### Field work

We captured 26 pregnant females of *L. pictus* by noose during two successive years: in early December 2022 and late November 2023, periods of the season during which embryos are expected to be at an early or intermediate stage of development (*sensu* Ibargüengoytía and Cussac, 1996). Captures were carried out along the shores of Lake Nahuel Huapi, San Carlos de Bariloche, Río Negro province, Argentina (41° 7.444′ S; 71° 21.606′ W 778 m asl). The capture location of each specimen was georeferenced to facilitate their subsequent return to the exact site of capture (GPS Garmin Map 60CSx). Immediately after capture, body temperature ($T_b$) was measured using a digital thermometer (TES 1303; TES Electrical Electronic Corp., Taipei, Taiwan, China, ±0.03°C) connected to a thermocouple (TES TP-K01, 1.62 mm diameter), which was inserted approximately 0.5 cm into the cloaca. We also recorded the substrate temperature ($T_s$) and the air temperature 1 cm above the ground ($T_a$) of the microenvironment where each lizard was captured.

### Ultrasound diagnosis and body measures

Lizards were individually placed in cloth bags and transported to the laboratory for pregnancy confirmation. Thus, pregnancy confirmation, the stage of embryonic development (expected for the experiments), and litter size were determined by ultrasound diagnosis. Diagnoses were made using a Chison Ultrasound Diagnostic System ECO1 VET PW, equipped with a 5.3-10 MHz linear transducer (Wuxi, Jiangsu, China). To do this, each specimen was gently placed on its back on the work table and held by hand to maintain its position (following Sacchi et al., 2012; Boretto et al., 2014; Bertocchi et al., 2018). Ultrasound gel was applied to the skin surface (approximately 0.5 cm thick) to facilitate continuous scanning and minimize interference with signal transmission during ventral and sagittal scans. We recorded the presence and number of embryos, the length and width of each embryo, and its echogenicity, as well as any informative observations to estimate the stage of pregnancy (initial, medium, or advanced; *sensu* Leyton et al., 1980). After ultrasound diagnosis of each female, the gel was carefully removed with paper towels, and body mass (Electronic Balance, ±0.01 g) and SVL (Wembley digital caliper, ±0.01 mm) were assessed. A total of 37 adult female lizards were captured; 26 of them exhibited early or intermediate stages of embryonic development, whereas 11 non-pregnant individuals were released the same day at their exact capture site. Pregnant females were then housed individually in open-top terrariums (115×18×30 cm; length×width×height) with sand from the capture site as substrate.

### Recording $T_{pref}$ of pregnant females

One day after arriving at the laboratory, the preferred body temperatures ($T_{pref}$1) were measured for all females. The $T_{pref}$ are the body temperatures selected by animals of a given species while thermoregulating under a full range of biologically relevant thermal conditions, typically in a laboratory thermal gradient free from external influences (Licht et al., 1966; Dawson, 1975; Huey, 1982). To achieve this, a thermal gradient of 20°C to 45°C was generated in each female terrarium using a 75 W incandescent lamp placed above one end of the terrarium. During this trial, female preferred body temperatures were recorded using an ultra-thin (0.076 mm; OMEGA) catheter thermocouple fastened to the abdomen and to the base of the tail with hypoallergenic tape, to keep the thermocouple in position during the experiment (following methods of Duran et al., 2020, 2023; Brizio et al., 2021). The thermocouples were connected to a temperature data acquisition module (USB-TC08; OMEGA, Biel/Bienne, Switzerland), which logged $T_{pref}$ every 2 s during 2.5 h. For each female, we estimated the mean $T_{pref}$, representing the average body temperature recorded throughout the entire 2.5-h thermoregulation trial, and the minimum and maximum $T_{set}$, which were represented by the 25th and 75th percentiles of the $T_{pref}$ data. The $T_{pref}$ of pregnant females was measured again 15 days after the start of the temperature experiments ($T_{pref}$2), as described in the following section. The $T_{pref}$ was not measured at the end of pregnancy to minimize stress during handling.

### Pregnant females experiment

During the experiments, females were fed once daily with *Tenebrio molitor* larvae supplemented with a reptile-specific multivitamin and calcium (ReptoCal, Tetrafauna™) and had access to water *ad libitum*. All females were maintained in the same room, under bright fluorescent light tubes (TDL 36 W/54 1) controlled by timers, which mimicked the daily light conditions of summer in a cycle of 14 h light:10 h dark, complemented with UVB light (Sylvania-Reptistar®; located 30 cm away from lizards; *sensu* Lindgren, 2004 and Fernández et al., 2017) during activity hours (from 9:00 AM to 6:00 PM). Ambient temperature was recorded every 5 min using data loggers (HOBO Onset Computer Corporation, Bourne, MA, USA) located inside the terrarium to control the mean temperature during the day and night.

The second day in laboratory, pregnant females were randomly assigned to one of the two temperature treatments until parturition: 1) NRG – during activity hours from 9:00 AM to 6:00 PM (*sensu* Kubisch et al., 2016a) females had free access to a thermoregulation area with a temperature gradient from 21°C to 45°C and free access to a shelter (20.5°C) in the cold extreme. Thus, pregnant females could freely thermoregulate and reach their $T_{pref}$ during activity hours (Fig. 2); 2) RG – had the same characteristics as NRG, but during the $h_r$, estimated based on predictions in a scenario of global warming (4.5 h – between 12:00 PM and 4:30 PM), pregnant females only had the option to expose themselves to a restriction temperature of 40.6°C or take refuge at 20.5°C (Fig. 2).

The restriction temperature of 40.6°C used in the RG was selected because it represents the maximum voluntary temperature registered in *L. pictus* and corresponds to the upper limit of its $T_{pref}$, serving as a threshold for thermal restriction (*sensu* Kubisch et al., 2016a). In *L. pictus*, limited plasticity in thermal biology has been demonstrated (Kubisch et al., 2016b), suggesting that its maximum voluntary temperature is unlikely to change in response to climate warming. In addition, the restriction period of 4.5 h ($h_r$) was estimated by adding a +3°C increase, based on climate change projections for the study area (IPCC, 2014; Barros et al., 2015), to operative temperature ($T_e$) data recorded during pregnancy months ($T_e$ data used and provided from Kubisch et al., 2016a). Thus, the $h_r$ reflect the daily hours during which temperatures, adjusted for future warming, are expected to exceed the restriction temperature of 40.6°C (*sensu* Kubisch et al., 2016a), thereby limiting activity. This approach provides a realistic estimate of potential activity restriction durations under extreme future thermal scenarios. Hence, the $h_r$ used in the RG were 4.5 h between hours of activity (median 4.33 h, range: 0 to 7.83 h).

The shelter in both treatments was 28×18×30 cm (length×width×height; Fig. 2) and had the same temperature as the average recorded in the shelters in their natural environment during activity hours (20.5°C; E.L.K., unpublished data). This temperature was achieved and maintained using an insulation system (expanded polystyrene walls) and thermal cooling of the surrounding environment around all shelters, implemented with a room air conditioner (Fig. S1).

The highest temperature of the thermal gradients of NRG and RG treatments was achieved with incandescent lamps (75 W) located 20 cm above one end of the terrarium. The restriction temperature (40.6°C) during the $h_r$ was maintained using an additional 75 W incandescent lamp

positioned 20 cm above and near the opposite end of the terrarium, close to the shelter (Fig. S1). Both treatments, NRG and RG, were conducted following the daily activity cycles of *L. pictus* observed in the field. At night, females remained at room temperature (16-20°C). Pregnant females that appeared weak, did not feed, or did not actively thermoregulate were removed from the experiment. After 20 days of experimentation, we conducted an additional ultrasound diagnosis on four females (two from each treatment, randomly selected) to confirm that embryonic development was progressing.

### Body condition and morphometric measures of females and newborns
Female body mass was reassessed on three additional occasions: 15 days after capture while still pregnant, immediately after parturition, and 10 days postpartum. Terrariums were checked daily for newborns. When a birth was recorded, the mother and her offspring were weighed and measured. Postpartum RG females were removed from the restriction treatment after parturition, allowing them to recover their body condition under the same thermal treatment as the NRG females. Reproductive effort was calculated as the mean litter mass/female mass after parturition (*sensu* Charnov, 2002). BCI of newborns and females was calculated as $\log_{10}$ mass/ $\log_{10}$ SVL (*sensu* Beltran et al., 2021, and Jiang et al., 2021). For all newborns, the SVL, head length, head width, inter-limb length, distance between knees, and tail length were measured using a digital caliper (Wembley ±0.01 mm; *sensu* Fernández et al., 2017). Only 'viable' newborns, fully developed and in apparently good health (*sensu* Lemus et al., 1981 and Fernández et al., 2017) were subjected to the locomotor performance experiment.

### Newborns' locomotor performance
Locomotor performance of *L. pictus* newborns was measured between 24 and 48 h after birth (following Angilletta et al., 2002; Fernández et al., 2017), as a relevant trait for assessing the organism's fitness (Wapstra, 2000; Fernández et al., 2017). One litter at a time was removed from the maternal terrarium without prior feeding and was placed in a quiet, dark environment before runs. To assess phenotype differences in the offspring, they were run at ambient temperature (21°C) and also at the mean preferred temperature of pregnant females ($T_{pref}1=34$°C) determined in this study. Following Fernández et al. (2017) methods, these temperatures were maintained using a heating system consisting of a thermal water bath (with controlled temperature and thermostat) and a fan with a hot-cold thermostat to keep the room at the desired experimental temperature. The $T_b$ of the newborns before the run was stabilized by placing each individual in an open-top cylinder container (7 cm diameter×12 cm height) inside a thermal water bath (±0.5°C, 6 liters) for 1 h before the run, allowing them to acclimatize to the temperature treatment.

Newborns were tested on a 1.15 m-long wooden racetrack with a cork substrate, a shelter at one end, and eight infrared photoreceptors spaced 0.15 m apart, connected to a computer. Running times between photoreceptors were used to calculate speed. The newborns were tested at both temperatures (21°C and 34°C, randomly assigned) on the same day, with at least 4 h of rest between sessions. Each newborn performed three runs at each temperature, with a 30-min resting period between runs. They were encouraged to run by gently touching their tails or the back of their legs, taking care not to interfere with their speed. For analysis, the fastest of the three runs for each lizard was considered their maximum velocity ($V_{max}$; *sensu* Fernández et al., 2017). Trials in which the newborn refused to run, stopped, or ran in the wrong direction were excluded. The speed during SR, defined as the fastest speed over the first 0.15 m of the track, was used to estimate acceleration and fear response, which are often seen in the field. Additionally, the speed for LR, the fastest speed over the full length of the racetrack (1.05 m), was calculated as a measure of the lizard's ability to maintain speed over a longer distance, reflecting their capacity for activities such as foraging, dispersal, and predator evasion. After assessing locomotor performance, the newborns were given food and water *ad libitum*.

Newborns were tested again 10 days after birth, having been maintained under the same temperature regimes as their mothers during pregnancy (RG or NRG, Fig. 2). Locomotor performance was reassessed at both test temperatures (21°C and 34°C, randomly assigned), and body condition was evaluated at the end of this period. Finally, each mother with their litter was released to their corresponding capture site, by using data of the capture sites previously geo-referenced (GPS GARMIN Map 60CSx) and photos of each microenvironment.

### Ethical considerations
Fieldwork was conducted under permits from the Administración de Parques Nacionales, Argentina (Permit number 1826). Procedures were approved by the Institutional Committee for the Care and Use of Laboratory or Experimental Animals (CICUAL) of INIBIOMA (Instituto de Investigaciones en Biodiversidad y Medioambiente) of CONICET, Argentina (protocol number 2020-032). Furthermore, the present study adhered to all pertinent guidelines and regulations, including the Guide for the Care and Use of Laboratory Animals (8th edition; National Academies Press, Washington, DC, USA) and Argentine National Law No. 14,346. No individuals were euthanized. Finally, the research adhered to the ARRIVE guidelines.

### Statistical analyses
We used the statistical software programs SigmaPlot 15.0® and SPSS 15.0®. Assumptions of normality and homogeneity of variance were checked using Shapiro-Wilk's and Levene's Test, respectively. Means were given as ±standard error (s.e.m.) and medians as 25th and 75th percentiles from the interquartile range.

We used *t*-test to analyse mean differences between females and offspring from NRG and RG treatments in the SVL, the reproductive effort, and body condition, and to determine the differences between the mean days from the start of the experiment to the day of parturition, as well as in the number of viable offspring, their morphometric variables, and locomotor performance. Paired *t*-test was used to analyse differences between pared sample means of the ultrasound litter size versus the litter size that females delivered, $T_{pref}$ of pregnant females when arrived to laboratory versus $T_{pref}$ after 15 days under experiment, BCI at the beginning of temperature treatments versus at birth for offspring and the next BCIs, and between locomotor speed in SR and LR at 21°C and 34°C, at birth and 10 days after. When normality and/or homogeneity of variance assumptions were not met, we employed the non-parametric Mann–Whitney *U*-test to compare medians between two independent samples, and the Wilcoxon signed-rank test for comparing medians of repeated samples across two trials. A binomial test was used to analyze the proportion of females who gave birth and completed the experiment compared to the total number of pregnant females. To assess interactions between two factors (treatment group and $T_{pref}$, or treatment group and BCI), we used a two-way repeated-measures ANOVA.

### Acknowledgements
We thank Dr Nora Ibargüengoytía for her insightful contributions at the beginning of this research. We also thank UNCOMA-CRUB and INIBIOMA-CONICET for providing us with the facilities including a laboratory to carry out the experiments. Research permits (N° 1826 from 2022 to 2024) were obtained from the Administración de Parques Nacionales, Argentina.

### Competing interests
The authors declare no competing or financial interests.

### Author contributions
Conceptualization: J.B.F., E.L.K., J.M.B.; Formal analysis: J.B.F., E.L.K., F.D., J.M.B.; Funding acquisition: J.B.F., J.M.B.; Investigation: J.B.F., E.L.K., F.D., J.M.B.; Methodology: J.B.F., E.L.K.; Project administration: J.B.F.; Resources: J.B.F.; Supervision: J.B.F., J.M.B.; Validation: J.B.F., E.L.K., F.D., J.M.B.; Visualization: J.B.F.; Writing – original draft: J.B.F., J.M.B.; Writing – review & editing: J.B.F., E.L.K., F.D., J.M.B.

### Funding
This research was funded by the Fondo para la Investigación Científica y Tecnológica (PICT-2020-SERIEA-3401; PICT-2020-SERIEA-03395), Universidad Nacional del Comahue (04/B234), and the Consejo Nacional de Investigaciones Científicas y Técnicas (CONICET, PIBAA 2023). Open Access funding provided by Instituto de Investigaciones en Biodiversidad y Medioambiente (INIBIOMA-CONICET). Deposited in PMC for immediate release.

**Data and resource availability**
All relevant data and details of resources can be found within the article and its supplementary information.

**Peer review history**
The peer review history is available online at https://journals.biologists.com/bio/lookup/doi/10.1242/bio.062159.reviewer-comments.pdf

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
