## [Peer Review File · Biology Open]

Giving Their All for Their Offspring: Physiological Trade-Offs in an Andean-Patagonian Viviparous Lizard in Response to Global Warming

Erika L. Kubisch, Fernando Duran, Jorgelina M. Boretto and Jimena B. Fernández

DOI: 10.1242/bio.062159

Editor: Lewis Halsey

Review timeline

Original submission:	7 July 2025
Editorial decision:	9 July 2025
Transfer to Biology Open:	10 July 2025
Editorial decision:	18 July 2025
Revision received:	25 August 2025
Accepted:	26 August 2025

Original submission

First decision letter

MS Title: Giving Their All for Their Offspring: Physiological Trade-Offs in an Andean- Patagonian Viviparous Lizard in Response to Global Warming

Authors: Jimena B Fernández; Erika L. Kubisch; Fernando Duran; Jorgelina M. Boretto

I have read, with much interest, the above manuscript, which you recently submitted to us. It is a very nice study and I can appreciate the amount of work that has gone into collecting the very large dataset. Unfortunately, however the focus of your paper is not really suitable for this journal. You very nicely describe pattern, but for this journal, there needs to be a stronger emphasis on the elucidation of mechanism through the measurement of physiological traits. There is just not enough physiology to warrant publication in this journal. I am really sorry to give you this disappointing news, but I am confident that your paper would not get through our reviewing process with strong enough reviews. Please do not be too disheartened by this outcome.

First decision letter

MS ID#: bio.062159

MS Title: Giving Their All for Their Offspring: Physiological Trade-Offs in an Andean- Patagonian Viviparous Lizard in Response to Global Warming

Authors: Jimena B Fernández; Erika L. Kubisch; Fernando Duran; Jorgelina M. Boretto

I have now reached a decision on the above manuscript.

The reviewer reports are shown at the bottom of this email or can be accessed, together with a copy of this decision letter, by going to:

As you will see, the reviewers raised a number of substantial criticisms that prevent me from accepting the paper at this stage.

They suggest, however, that a revised version might prove acceptable, if you can address their concerns. If you think that you can deal satisfactorily with the criticisms on revision, I would be pleased to see a revised manuscript. We would then return it to the reviewers.

At this stage, we also ask you to ensure your manuscript complies with our formatting guidelines. Provided you are able to fully address the referees' comments, we are positive about publication of your paper (we accept over 95% of revision submissions) and therefore hope you won't mind any extra work involved in reformatting your manuscript at this point.

Please ensure that you clearly highlight all changes made in the revised manuscript. Please avoid using 'Tracked changes' in Word files as these are lost in PDF conversion.

I should be grateful if you would also provide a point-by-point response detailing how you have dealt with the points raised by the reviewers in the 'Response to Reviewers' box. Please attend to all of the reviewers' comments. If you do not agree with any of their criticisms or suggestions please explain clearly why this is so.

Reviewer 1

Comments to Author

The authors conducted an experimental study, whereby captured wild-type pregnant lizards (*Liolaemus pictus*) were randomly maintained at one of two different thermal conditions, i.e., typical ambient temperatures in the region, and projected increases by the end of the century due to global warming. They reported lower preferred body temperatures and stagnant body condition in the group exposed to the higher temperatures, than those in the group maintained at ambient temperatures, but no influence on offspring body condition or locomotion. They concluded that the pregnant females physiologically compensated for the negative effects of the thermal manipulation to support embryo development, as reflected in the title of "Giving Their All for Their Offspring".

Overall, I enjoyed this manuscript and considered it to be well-written, to address an important question and to employ an appropriate design. There were, however, some elements of the methods that I felt lacked clarity, which may have impacted my interpretation of the results and the authors conclusions. Furthermore, some clarification on the statistical approach employed is important as I believe that certain findings may potentially be less conclusive than implied within the discussion. My specific suggestions and queries are listed below.

1. I suggest that the authors reconsider the term "activity restriction" when describe their intervention. In the abstract and early sections, the term could be taken to imply that physical constraints were imposed, however it becomes clear on lines 234 - 238 that activity was not physically limited, but that the temperatures meant that the females only had the option of seeking refuge at a lower temperature, or exposing themselves to the higher one. I believe this to be an important distinction. The introduction (e.g., on lines 87 - 93) indicates that restricted activity, which may impact foraging and social behavior, is a likely consequence of rising temperature, however this appears to be a hypothesis rather than a certainty (line 142). As such, it may be more accurate to describe the experiment in terms of temperature manipulation or restriction, with activity restriction presented as a likely outcome of this intervention, rather than the intervention itself. Are any activity data available? Did the RG consistently seek refuge at 20.5°C during Rh, making them less active overall than NRG?
2. Lines 28 - 32: I suggest you provide the total numbers randomized to each group in the abstract, as this would allow greater contextualization of the % loss data reported.
3. Line 50: Delete "the" from the sentence "with the record-high temperatures";

4. Line 51: Consider including some examples of which specific hydrological variables you are referring to.
5. Lines 56 - 58: The cited sources for projected temperature increases are over a decade old, and if possible, it would be useful to use more recent projections, and to consider whether these past projections are consistent with observed temperature trends over the past decade.
6. Line 145: Gestation time was indicated as a prediction here, but not reported within the results. Lines 400 - 401 indicates no difference in the number of days from the start of the experiment to the day of parturition between groups, but does this necessarily represent gestation time, considering that there may have been some variation in the pregnancy stage at the beginning of the experiment?
7. Line 159: Considering that the concept of preferred temperature is core to the study, I suggest that you clearly define it here. Furthermore, it was not entirely clear (at least to me!) how T_{pref} was actually recorded in the description provided on lines 202 - 216. Detailed information about the thermal gradient, and body temperature measurement were provided, but how was actual T_{pref} determined? Was it the temperature at which the lizards chose to spend most time in? Or the temperature at which they were most active? Or as the mean recorded temperature across the entire 2.5h collection?
8. Field-work: Did you do some on-site assessment to ensure that they were early-stage pregnant females before taking them to the lab, or did you take all captured lizards to the lab for assessment? How many lizards were actually captured and at what point were non-pregnant females or males released?
9. Line 199: Typo here - should be "assessed".
10. Line 269: What was the purpose of the repeated ultrasound diagnosis of 4 females after 20 days? If confirmation of embryonic development was important to the study, why was this not conducted on all lizards? Were the 4 females randomly selected?
11. Locomotor performance test: How was the test started? Was there some stimulus to encourage the baby lizards to run at their maximum speed along the racetrack, e.g., a noise or movement to frighten them into movement? Did each newborn complete 3 trials at each temperature condition? If so, it would be useful to indicate this on Line 306.
12. Statistical analysis: Please clarify when the repeated measured ANOVA was employed. Considering that the primary question was to compare outcomes between the two experimental conditions, it would appear to be more appropriate to conduct a repeated measures ANOVA for all outcomes for which repeated measures were available, rather than conducting paired sample t-tests on individual groups. This is particularly relevant for the body condition outcome. The main conclusion of the study is based on an assumed BCI stability in RG compared to an increase in NRG, however a statistical difference in one group but not the other does not necessarily mean that there was a difference between the groups. I believe that testing for an interaction between group and BCI change may be a more appropriate and rigorous test than conducting repeated paired sample tests.
13. Line 437 - 438: The finding of reduced BCI of NRG newborns was interesting, and likely warrants some consideration in the discussion. What are the authors thoughts on this finding?
14. Lines 468 - 480: In my opinion, the finding of a greater proportion of lizards removed from the experiment due to weakness or non-feeding in the RG and NRG group is notable, and implies that the temperature manipulation impacted pregnancy viability. As such, I suggest commenting on it within the opening paragraph of the discussion when summarizing key findings.
15. Line 503 - 505: I am not entirely clear on how the reduction of preferred temperature observed in the RG suggests a strategy to enhance offspring fitness. Considering that this is a key finding, further elaboration may be useful and interesting.

16. Line 521 - 529: In my opinion, the conclusions reached here seem somewhat strong, when considered in light of the reported results and in my opinion, some tempering and a more balanced critique of study findings would be useful. As described above, it is not entirely clear if the RG really did differ in their BCI development compared to NRG, considering the statistical analysis employed did not allow for this interpretation. Figure 4 appears to indicate similar trends and substantial overlap between both groups. It would be interesting to plot individual data in this figure to identify if the apparently higher BCI in NRG at time point 2 reflected the group as a whole, or if there were any extreme data points in either groups that may have influenced this. Furthermore, and as stated on 491, the RG did exhibit significant mass recovery quickly after parturition. As such, it seems quite speculative to predict that lizards might skip more years of reproduction based on these findings. Finally, on line 157, the species are already described as biennial or triennial breeders, and as such to have substantial recovery time between reproductive cycles.

Reviewer 2

Comments to Author

I appreciate the authors' time and effort for the manuscript. This study investigates how projected climate warming and associated behavioral thermoregulation constraints affect reproductive success and offspring quality in the viviparous lizard *Liolaemus pictus*. Using a robust laboratory experiment with two thermal scenarios—No Restriction Group (NRG) and Restriction Group (RG)—the authors demonstrate that although activity restrictions negatively impact maternal condition and feeding behavior, females appear to physiologically compensate to ensure offspring fitness remains unaffected. This work contributes to our understanding of thermal plasticity in viviparous reptiles and their potential resilience to climate change.

The manuscript presents a well-designed and ethically sound experimental study with strong ecological relevance. The authors effectively combine physiological measurements, thermal preference assessments, and behavioral trials to evaluate the impact of climate-related thermal restriction. The results are clearly presented and well-supported by statistical analyses. The discussion is appropriately nuanced, addressing both the resilience and limitations of physiological plasticity.

The work is suitable for publication pending minor to moderate revisions, particularly regarding presentation clarity, experimental detail, and improvement of English language in some sections.

Comments:

1. While the manuscript is mostly readable, several sentences require editing for grammar and clarity.
2. The restriction temperature and restriction period duration should be more clearly justified in the Methods. The rationale for selecting 40.6°C as the threshold and 4.5 hours of restriction is mentioned but should be synthesized more concisely.
3. A schematic of the thermal gradient setup and restriction protocol (e.g., a simplified version of Fig. 2) in the supplementary material would aid comprehension.
4. The authors touch upon the potential cost of repeated reproductive trade-offs under chronic warming (e.g., biennial reproduction). This is an important point that could be expanded in discussion to suggest specific future research directions.
5. Figure captions should be more self-contained and detailed.
6. Table 1, 2, and 5 is repetitive with Fig 4, 5, and 6. Tables can be put into supplementary or only the stat results can be presented in one table.
7. Line 422-423: n value needs to be fixed.
8. Line 458: Figure cited wrongly (Fig 3 is cited instead of Fig 6).

Reviewer's Responses to Questions

Experimental quality

Does each figure have the proper controls?

If 'No', please indicate reasons in Comments for Author box below.

Reviewer #1:

- Yes

Reviewer #2:

- Yes

Were the data analyzed using appropriate statistical tests?

If 'No', please indicate reasons in Comments for Author box below.

Reviewer #1:

- No

Reviewer #2:

- Yes

Reproducibility

Were experiments performed using adequate number of biological replicates?

If 'No', please indicate reasons in Comments for Author box below.

Reviewer #1:

- Yes

Reviewer #2:

- Yes

Does the methods section provide sufficient detail to permit reproducibility?

If 'No', please indicate reasons in Comments for Author box below.

Reviewer #1:

- Yes

Reviewer #2:

- Yes

Completeness

Are the manuscript's conclusions supported by the data?

If 'No', please indicate reasons in Comments for Author box below.

Reviewer #1:

- Yes

Reviewer #2:

- No

Scholarship

Do the authors cite and discuss the merits of data that would argue for and against their conclusion?

If 'No', please indicate reasons in Comments for Author box below.

Reviewer #1:

- Yes

Reviewer #2:

- No

Does the manuscript title & abstract accurately reflect the contents of the manuscript, without hyperbole?

If 'No', please indicate reasons in Comments for Author box below.

Reviewer #1:

- Yes

Reviewer #2:

- Yes

First revision

Author response to reviewers' comments

Handling Editor Biology Open

Dear Dr. Halsey,

We would like to thank the Reviewers for their thorough and constructive evaluation of our manuscript. We have carefully considered all comments and suggestions and have revised the manuscript accordingly. Below, we provided a detailed point-by-point response to the Reviewers' suggestions, with our replies highlighted in bold. Additionally, following the "Open Biology Formatting guidelines", we reduced the Abstract to no more than 200 words. All changes made in the revised manuscript file have been highlighted in yellow.

Reviewer 1: The authors conducted an experimental study, whereby captured wild-type pregnant lizards (*liolaemus pictus*) were randomly maintained at one of two different thermal conditions, i.e., typical ambient temperatures in the region, and projected increases by the end of the century due to global warming. They reported lower preferred body temperatures and stagnant body condition in the group exposed to the higher temperatures, than those in the group maintained at ambient temperatures, but no influence on offspring body condition or locomotion. They concluded that the pregnant females physiologically compensated for the negative effects of the thermal manipulation to support embryo development, as reflected in the title of "Giving Their All for Their Offspring".

Overall, I enjoyed this manuscript and considered it to be well-written, to address an important question and to employ an appropriate design. There were, however, some elements of the methods that I felt lacked clarity, which may have impacted my interpretation of the results and the authors conclusions. Furthermore, some clarification on the statistical approach employed is important as I believe that certain findings may potentially be less conclusive than implied within the discussion. My specific suggestions and queries are listed below.

1. I suggest that the authors reconsider the term "activity restriction" when describe their intervention. In the abstract and early sections, the term could be taken to imply that physical constraints were imposed, however it becomes clear on lines 234 - 238 that activity was not physically limited, but that the temperatures meant that the females only had the option of seeking refuge at a lower temperature, or exposing themselves to the higher one. I believe this to be an important distinction. The introduction (e.g., on lines 87 - 93) indicates that restricted activity, which may impact foraging and social behavior, is a likely consequence of rising temperature, however this appears to be a hypothesis rather than a certainty (line 142). As such, it may be more accurate to describe the experiment in terms of temperature manipulation or restriction, with activity restriction presented as a likely outcome of this intervention, rather than the intervention itself. Are any activity data available? Did the RG consistently seek refuge at 20.5°C during Rh, making them less active overall than NRG?

According to the Reviewer's suggestion, we have reviewed the use of the terminology "activity restriction" throughout the manuscript. Now, we consistently refer to this restriction as "thermal restriction" to more accurately reflect that the manipulation was made on the thermal environment, and was not a physical limitation of movement.

Furthermore, although it would have been ideal to use video recording to quantify activity levels during the restriction period, the experimental design was precisely adjusted before

implementation. We conducted visual checks and consistently confirmed that all lizards in the thermal restriction group remained inside the shelters during the entire restriction period.

2. Lines 28 - 32: I suggest you provide the total numbers randomized to each group in the abstract, as this would allow greater contextualization of the % loss data reported.

Following the Reviewer's suggestion, we have included the total numbers randomized to each group in the Abstract (lines 10 and 11).

3. Line 50: Delete "the" from the sentence "with the record-high temperatures..."

Done

4. Line 51: Consider including some examples of which specific hydrological variables you are referring to.

Following the Reviewer's suggestion, we added examples of which hydrological variables we are referring to in that sentence (lines 26-27).

5. Lines 56 - 58: The cited sources for projected temperature increases are over a decade old, and if possible, it would be useful to use more recent projections, and to consider whether these past projections are consistent with observed temperature trends over the past decade.

The experimental work for this study began in 2022, and the experimental design was therefore based on the climate projections available at that time, specifically those provided in the IPCC Fifth Assessment Report (AR5, 2014), as noted in the Materials and Methods section. The subsequent IPCC Sixth Assessment (AR6) Synthesis Report, released in 2023, also projects an increase in ambient temperature exceeding 3 °C by 2070 under an extreme emissions scenario (RCP 8.5), for northwestern Patagonia. We have now incorporated a citation to the AR6 report in the Introduction (lines 32-34) to reflect the most recent projections.

6. Line 145: Gestation time was indicated as a prediction here, but not reported within the results. Lines 400 - 401 indicates no difference in the number of days from the start of the experiment to the day of parturition between groups, but does this necessarily represent gestation time, considering that there may have been some variation in the pregnancy stage at the beginning of the experiment?

According to previous studies (Ibargüengoytia & Cussac, 1996) and our field observations (from emergence after hibernation to the first sighting of pregnant females), reproductive timing among pregnant females is largely synchronized. Nevertheless, to avoid potential confusion, we have removed this prediction from the manuscript and kept the focus on the expected effects on the offspring.

7. Line 159: Considering that the concept of preferred temperature is core to the study, I suggest that you clearly define it here. Furthermore, it was not entirely clear (at least to me!) how T_{pref} was actually recorded in the description provided on lines 202 - 216. Detailed information about the thermal gradient, and body temperature measurement were provided, but how was actual T_{pref} determined? Was it the temperature at which the lizards chose to spend most time in? Or the temperature at which they were most active? Or as the mean recorded temperature across the entire 2.5h collection?

Following the reviewer's suggestion, we removed the reference to preferred temperature from this line, as it was not the appropriate place to elaborate on the concept. We have now provided a clear and precise definition of T_{pref} in the "Recording T_{pref} of pregnant females" section (lines 187-190).

In this study, T_{pref} was recorded for each female (within its terrarium) using an ultra-thin thermocouple connected to a Data Acquisition Module, which measured body temperature every 2 s over a 2.5-h period. For each female, we calculated the mean

T_{pref} as the average body temperature recorded throughout the entire 2.5-h thermoregulation trial. This clarification has been added between lines 192-201.

8. Field-work: Did you do some on-site assessment to ensure that they were early- stage pregnant females before taking them to the lab, or did you take all captured lizards to the lab for assessment? How many lizards were actually captured and at what point were non-pregnant females or males released?

Adult females and males of *Liolaemus pictus* are easily identified in the field. Males, unlike females, possess orange pre-cloacal pores, larger head length and width, and a wider tail base width; the latter being the most reliable field trait for sex identification prior to capture. Additionally, they reach sexual maturity at a standard length of 49 mm SVL. This information about the species was included between lines 132-134.

After visual confirmation in the field, all captured adult females were transported to the laboratory, located approximately 30 minutes from the field site. There, we performed detailed ultrasound examinations to confirm pregnancy and determine the stage of embryonic development. In total, 37 adult female lizards were captured; 26 had an early or intermediate stage of embryo development, and 11 non-pregnant individuals were released the same day at their exact capture site. This information was included between lines 179-181.

9. Line 199: Typo here - should be "assessed".

Done.

10. Line 269: What was the purpose of the repeated ultrasound diagnosis of 4 females after 20 days? If confirmation of embryonic development was important to the study, why was this not conducted on all lizards? Were the 4 females randomly selected?

An additional ultrasound diagnosis was performed after 20 days to confirm that embryonic development was progressing. This procedure was conducted on only four randomly selected females (two from each treatment; information included in line 257), and the resulting data were used as our reference (illustrated in Fig.3C). To minimize handling stress, we did not perform ultrasound diagnosis on all females.

11. Locomotor performance test: How was the test started? Was there some stimulus to encourage the baby lizards to run at their maximum speed along the racetrack, e.g., a noise or movement to frighten them into movement? Did each newborn complete 3 trials at each temperature condition? If so, it would be useful to indicate this on Line 306.

Following the Reviewer's suggestion, we indicate that newborns did three runs at each temperature condition on line 298.

In addition, newborns were encouraged to run by gently touching their tails or the back of their legs, taking care not to interfere with their running speed (following Fernández et al., 2017). This information was added between lines 297- 298, and the entire paragraph was rewritten to improve clarity.

12. Statistical analysis: Please clarify when the repeated measured ANOVA was employed. Considering that the primary question was to compare outcomes between the two experimental conditions, it would appear to be more appropriate to conduct a repeated measures ANOVA for all outcomes for which repeated measures were available, rather than conducting paired sample t-tests on individual groups. This is particularly relevant for the body condition outcome. The main conclusion of the study is based on an assumed BCI stability in RG compared to an increase in NRG, however a statistical difference in one group but not the other does not necessarily mean that there was a difference between the groups. I believe that testing for an interaction between group and BCI change may be a more appropriate and rigorous test than conducting repeated paired sample tests.

We agree with the Reviewer, and we now use a two-way RM ANOVA to compare outcomes between the two experimental conditions. We employed this statistical analysis to compare groups (RG and NRG) at two different times in females: BCI₁ vs. BCI₂ (lines 371-373), BCI₃ vs. BCI₄ (lines 396-398), and T_{pref1} vs. T_{pref2} (line 405); and in their offspring: BCI_{birth} vs. BCI_{10-d} (lines 426-430).

13. Line 437 - 438: The finding of reduced BCI of NRG newborns was interesting, and likely warrants some consideration in the discussion. What are the authors thoughts on this finding?

We agree with the Reviewer that the finding of reduced BCI of NRG newborns was interesting. We included this issue in the "Discussion" section (lines 483- 499).

14. Lines 468 - 480: In my opinion, the finding of a greater proportion of lizards removed from the experiment due to weakness or non-feeding in the RG and NRG group is notable, and implies that the temperature manipulation impacted pregnancy viability. As such, I suggest commenting on it within the opening paragraph of the discussion when summarizing key findings.

According to the Reviewer's suggestion, we have included in the opening paragraph of the Discussion the information on the proportion of lizards that had to be removed from the experiment due to cessation of feeding or spontaneous abortions, reflecting not only the impact of thermal restriction on pregnancy viability in the RG, but also the effects of captivity-induced stress in both groups (Lines 470-473).

15. Line 503 - 505: I am not entirely clear on how the reduction of preferred temperature observed in the RG suggests a strategy to enhance offspring fitness. Considering that this is a key finding, further elaboration may be useful and interesting.

We have strengthened the discussion of these results to highlight the possible strategy of RG females to select lower and more stable preferred temperatures during pregnancy in a thermally restricted environment as a physiological compensation strategy to maintain offspring fitness stability while maintaining stable body condition (lines 500-527).

16. Line 521 - 529: In my opinion, the conclusions reached here seem somewhat strong, when considered in light of the reported results and in my opinion, some tempering and a more balanced critique of study findings would be useful. As described above, it is not entirely clear if the RG really did differ in their BCI development compared to NRG, considering the statistical analysis employed did not allow for this interpretation. Figure 4 appears to indicate similar trends and substantial overlap between both groups. It would be interesting to plot individual data in this figure to identify if the apparently higher BCI in NRG at time point 2 reflected the group as a whole, or if there were any extreme data points in either groups that may have influenced this. Furthermore, and as stated on 491, the RG did exhibit significant mass recovery quickly after parturition. As such, it seems quite speculative to predict that lizards might skip more years of reproduction based on these findings. Finally, on line 157, the species are already described as biennial or triennial breeders, and as such to have substantial recovery time between reproductive cycles.

We agree with the Reviewer.

We changed this paragraph of the Discussion based on the analysis results suggested by both Reviewers (lines 528-542).

In addition, we detected confusion related to the BCI₄ of postpartum females.

The purpose of comparing BCI₃ and BCI₄ in both groups was to monitor the post- parturition recovery of females' body condition and ensure their release in optimal condition. However, we agree with the Reviewer that it was not well explained and used throughout the manuscript.

Since postpartum RG females were removed from the restriction treatment after parturition (allowing them to recover their body condition under thermal conditions similar to those of the NRG females), the way to show data regarding BCI₄ in Fig. 4 was now modified to avoid confusion (using a black dashed line; see Fig. 4). The post-parturition procedure for RG females is now clarified in the “Methodology” section (lines 264-266), with particular emphasis on avoiding any comparison of BCI₄ between groups in the “Results” section.

It is likely that RG females, if they continued to experience hours of activity restriction due to high temperatures after parturition, would have died or, at the very least, failed to recover their body condition sufficiently for safe return to their natural habitat. This possibility, however, could be addressed in future studies (as discussed between lines 528-553).

Reviewer 2: I appreciate the authors' time and effort for the manuscript. This study investigates how projected climate warming and associated behavioral thermoregulation constraints affect reproductive success and offspring quality in the viviparous lizard *Liolaemus pictus*. Using a robust laboratory experiment with two thermal scenarios—No Restriction Group (NRG) and Restriction Group (RG)—the authors demonstrate that although activity restrictions negatively impact maternal condition and feeding behavior, females appear to physiologically compensate to ensure offspring fitness remains unaffected. This work contributes to our understanding of thermal plasticity in viviparous reptiles and their potential resilience to climate change.

The manuscript presents a well-designed and ethically sound experimental study with strong ecological relevance. The authors effectively combine physiological measurements, thermal preference assessments, and behavioral trials to evaluate the impact of climate-related thermal restriction. The results are clearly presented and well-supported by statistical analyses. The discussion is appropriately nuanced, addressing both the resilience and limitations of physiological plasticity.

The work is suitable for publication pending minor to moderate revisions, particularly regarding presentation clarity, experimental detail, and improvement of English language in some sections.

Comments:

1. While the manuscript is mostly readable, several sentences require editing for grammar and clarity.

According to the Reviewer's suggestion, we have thoroughly reviewed the English language throughout the manuscript. Please let us know if any specific paragraphs or sentences still require further clarification or grammatical improvement.

2. The restriction temperature and restriction period duration should be more clearly justified in the Methods. The rationale for selecting 40.6 °C as the threshold and 4.5 hours of restriction is mentioned but should be synthesized more concisely.

Following the Reviewer's suggestion, we have rewritten the paragraph regarding the restriction temperature and hours of restriction to provide a clearer and more concise justification of these methods (see lines 227-241).

3. A schematic of the thermal gradient setup and restriction protocol (e.g., a simplified version of Fig. 2) in the supplementary material would aid comprehension.

Following the Reviewer's suggestion, we have included a representative diagram of the temperature treatments and their thermal surroundings in the supplementary material (now labeled as Figure S1), to aid in the comprehension of the thermal gradient setup and the thermal restriction protocol.

4. The authors touch upon the potential cost of repeated reproductive trade-offs under chronic warming (e.g., biennial reproduction). This is an important point that could be expanded in discussion to suggest specific future research directions.

We agree with the Reviewer. This point was expanded in the Discussion, and we explain our future studies about this issue (lines 528-553).

5. Figure captions should be more self-contained and detailed.

Following the Reviewer's comment, we rewrote all Figure legends to be more self-contained and detailed.

6. Table 1, 2, and 5 is repetitive with Fig 4, 5, and 6. Tables can be put into supplementary or only the stat results can be presented in one table.

Following the Reviewer's suggestion, some of the statistical results from the former Table 1 have been incorporated into the text (line 357; and lines 369-377), and the Table itself has been removed. The previous Table 2 has been renumbered as Table 1, and Table 5 has been moved to the supplementary material as "Table S1". Consequently, the former Table S1 has now been relabeled as "Table S2".

7. Line 422-423: *n* value needs to be fixed.

In those lines (now labeled as lines 413 and 414), the reported *n* values correspond to the preferred body temperature data points recorded for each female during her entire parturition period in the preferred temperature experiment (one T_{pref} value recorded each 2 seconds).

8. Line 458: Figure cited wrongly (Fig 3 is cited instead of Fig 6).

Corrected.

Third decision letter

MS ID#: bio.062159R1

MS Title: Giving Their All for Their Offspring: Physiological Trade-Offs in an Andean- Patagonian Viviparous Lizard in Response to Global Warming

Authors: Jimena B Fernández; Erika L. Kubisch; Fernando Duran; Jorgelina M. Boretto

I have had the opportunity to read through both your rebuttal and the associated edits within your resubmitted manuscript. I see that you have attended to the Reviewers' concerns well and so I am now happy to tell you that your manuscript has been accepted for publication in Biology Open, pending our standard publication integrity checks. It was accepted on 26th August 2025.